# Physical activity prevents acute inflammation in a gout model by downregulation of TLR2 on circulating neutrophils as well as inhibition of serum CXCL1 and is associated with decreased pain and inflammation in gout patients

Kyle Jablonski[1‡], Nicholas A. Young[1‡], Caitlin Henry[1], Kyle Caution[2], Anuradha Kalyanasundaram[3], Ifeoma Okafor[1], Peter Harb[1], Emmy Schwarz[1], Paul Consiglio[3], Chris M. Cirimotich[4], Anna Bratasz[5], Anasuya Sarkar[3], Amal O. Amer[2], Wael N. Jarjour[1], Naomi Schlesinger[6‡]*

1 Division of Immunology and Rheumatology, Department of Internal Medicine, The Ohio State University Wexner Medical Center, Columbus, OH, Unites States of America, 2 Department of Microbial Infection and Immunity, The Ohio State University Wexner Medical Center, Columbus, OH, United States of America, 3 Department Physiology and Cell Biology, The Ohio State University, Columbus, OH, United States of America, 4 Battelle Biomedical Research Center, West Jefferson, OH, United States of America, 5 Small Animal Imaging Core, The Ohio State University Wexner Medical Center, Columbus, OH, United States of America, 6 Division of Rheumatology, Department of Medicine, Rutgers Robert Wood Johnson Medical School, New Brunswick, NJ, United States of America

‡ KJ and NAY share first authorship on this work. NAY and NS are joint senior authors on this work.
* schlesna@rwjms.rutgers.edu

## Abstract

### Objectives

Gout is the most prevalent inflammatory arthritis. To study the effects of regular physical activity and exercise intensity on inflammation and clinical outcome, we examined inflammatory pathogenesis in an acute model of murine gout and analyzed human gout patient clinical data as a function of physical activity.

### Methods

NF-κB-luciferase reporter mice were organized into four groups and exercised at 0 m/min (non-exercise), 8 m/min (low-intensity), 11 m/min (moderate-intensity), and 15 m/min (high-intensity) for two weeks. Mice subsequently received intra-articular monosodium urate (MSU) crystal injections (0.5mg) and the inflammatory response was analyzed 15 hours later. Ankle swelling, NF-κB activity, histopathology, and tissue infiltration by macrophages and neutrophils were measured. Toll-like receptor (TLR)2 was quantified on peripheral monocytes/neutrophils by flow cytometry and both cytokines and chemokines were measured in serum or synovial aspirates. Clinical data and questionnaires accessing overall physical activity levels were collected from gout patients.

**Data Availability Statement:** All relevant data are within the paper and its Supporting Information files.

**Funding:** Both Nicholas Young and Naomi Schlesinger had research support initially derived from Ardea Biosciences (this support was subsequently provided via Ironwood Pharmaceuticals) examining the role of exercise in inflammation, which directly supported these research endeavors. The Center for Clinical and Translational Science (CCTS) at OSUWMC is supported by UL1TR001070 from the National Center for Advancing Translational Science. The Comparative Pathology & Mouse Phenotyping Shared Resource at The Ohio State University is supported in part by grant P30 CA016058, National Cancer Institute. The funders had no role in study design, data collection and analysis, decision to publish, or preparation of the manuscript.

**Competing interests:** Naomi Schlesinger has grants from Pfizer and AMGEN; she is also on the advisory board and consulting for Novartis, Horizon Pharma, Selecta Biosciences, Olatec, and Mallinckrodt Pharmaceuticals. This does not alter our adherence to PLOS ONE policies on sharing data and materials. The remaining authors declare no conflict of interest. Ardea Biosciences (this support was subsequently provided via Ironwood Pharmaceuticals) funded this study, but did not contribute to the preparation of the manuscript and had no competing interests in this study. This does not alter our adherence to PLOS ONE policies on sharing data and materials.

## Results

Injection of MSU crystals produced a robust inflammatory response with increased ankle swelling, NF-κB activity, and synovial infiltration by macrophages and neutrophils. These effects were partially mitigated by low and moderate-intensity exercise. Furthermore, IL-1β was decreased at the site of MSU crystal injection, TLR2 expression on peripheral neutrophils was downregulated, and expression of CXCL1 in serum was suppressed with low and moderate-intensity exercise. Conversely, the high-intensity exercise group closely resembled the non-exercised control group by nearly all metrics of inflammation measured in this study. Physically active gout patients had significantly less flares/yr, decreased C-reactive protein (CRP) levels, and lower pain scores relative to physically inactive patients.

## Conclusions

Regular, moderate physical activity can produce a quantifiable anti-inflammatory effect capable of partially mitigating the pathologic response induced by intra-articular MSU crystals by downregulating TLR2 expression on circulating neutrophils and suppressing systemic CXCL1. Low and moderate-intensity exercise produces this anti-inflammatory effect to varying degrees, while high-intensity exercise provides no significant difference in inflammation compared to non-exercising controls. Consistent with the animal model, gout patients with higher levels of physical activity have more favorable prognostic data. Collectively, these data suggest the need for further research and may be the foundation to a future paradigm-shift in conventional exercise recommendations provided by Rheumatologists to gout patients.

## Introduction

Gout is a type of inflammatory arthritis characterized by monosodium urate (MSU) crystal-induced inflammation in joints, tendons, and surrounding tissues. Overconsumption of purine rich food/drinks and various health factors (obesity, heart or kidney disease) can lead to elevated serum uric acid and form MSU crystal deposits in the body. Untreated, gout can cause irreversible joint damage, chronic pain, and deformation. In addition to dietary and lifestyle changes, pharmacological treatments for the management of gout include a combination of anti-inflammatory and urate reducing agents.

Numerous epidemiological studies have shown that regular exercise produces an immuno-modulatory effect, which may contribute to the associated health benefits observed [1–3]. In the context of rheumatic diseases, exercise has been shown to be anti-inflammatory and capable of improving overall health by decreasing the incidence of co-morbidities [4].

Despite the high prevalence of gout and evidence underscoring the benefits of physical activity in rheumatic disease [5], there has been little investigation into the relationship between regular exercise and inflammation in the context of gouty arthritis. Consequently, there were no exercise guidelines for gout patients in the clinical practice recommendations by the American College of Rheumatology (2012) [6] and the American College of Physicians (2016) [7].

This study investigates the effects of physical activity in a murine model of acute gout and shows that daily aerobic exercise at both low and moderate-intensity levels can partially

mitigate the inflammatory response induced by intra-articular MSU crystals. These data indicate that reduced TLR2 expression in peripheral neutrophils and suppressed serum CXCL1 expression lead to a decreased inflammatory response. Therefore, the plausible mechanisms contributing to exercise-mediated suppression of NF-κB activity and IL-1β expression locally can be explained, at least in part, by a reduction of TLR2 expression on circulating neutrophils and a down-regulation of systemic CXCL1 expression. In concordance, physically active gout patients had significantly less flares/yr, decreased C-reactive protein (CRP) levels, and lower pain scores.

## Methods

### Mice

BALB/c mice were purchased from the Jackson Laboratories (Bar Harbor, ME) and BALB/C-Tg(NFκB-RE-luc)-Xen mice were purchased from Caliper Life Science (Hopkinton, MA). All animals were housed at The Ohio State University Wexner Medical Center (OSUWMC) in a BSL-2 barrier facility; maintenance and protocols were specifically approved for this study by the Institutional Animal Care and Use Committee (IACUC) through The University Laboratory Animal Resources at OSUWMC. The animal facility was maintained at 22–23˚C and 30–50% relative humidity with a 12-hour light/dark cycle; chow and water were supplied ad libitum. Euthanasia was performed by cervical dislocation under anesthesia using IACUC-approved procedures described in the associated animal protocol for this study.

### Respirometry

Exercise intensity levels were determined using a dual-chamber metabolic modular treadmill (Columbus Instruments, Model 1012M-2) capable of quantifying cardiopulmonary oxygen consumption rates during exercise. Male and female Balb/c mice of various ages from eight weeks to eight months (n = 14) were exercised in five min intervals at increasing speeds with a 5% incline. $VO_2$ consumption was recorded every 30 seconds. Values were plotted and a polynomial curve was applied to interpolate speeds on a graph correlating with 35% (low-intensity), 55% (medium-intensity), and 75% (high-intensity) $VO_2$ max.

### Exercise regimen

Male and female mice were exercised on an Exer3/6 treadmill and controller (Columbus Instruments, Columbus, OH). Mice were gradually acclimated to target speeds over a period of two to four weeks by incrementally increasing speed and duration until target conditions were reached. Mice were subsequently exercised daily for two weeks at 8 m/min, 11 m/min, or 15 m/min for 45 minutes at 5% incline. Control mice (0 m/min) were handled similarly by placing on non-moving treadmill and received no exercise.

### Induction of a gout glare and measurement of ankle diameter

An acute model of gout was generated by injecting MSU crystals into the intra-articular space of the right ankle of male Balb/C or BALB/C-Tg(NFκB-RE-luc)-Xen mice ranging from eight weeks to twenty weeks of age using a previously-characterized model of MSU crystal-induced ankle arthritis [8]. Approximately two hrs post-exercise, mice were sedated and baseline caliper (Fischer Brand) measurements of ankle diameter were recorded. Mice subsequently received a 20 μg injection of solubilized MSU crystals (Invivogen) administered intra-articularly. Changes in ankle diameter were determined by subtracting the initial caliper measurement from the 15-hr post-injection caliper measurement.

## *In vivo* imaging

BALB/C-Tg(NFκB-RE-luc)-Xen mice were imaged as described previously [9]. Briefly, mice were given 150 mg/kg luciferin [(Gold Biotechnology, Inc.); 15 mg/mL in PBS (pH 7; unadjusted)] through intraperitoneal injection and bioluminescent signals were captured following indicated time periods using the IVIS Lumina II (Xenogen). Data were quantitatively analyzed using IVIS Living Image software (v4.5; PerkinElmer). Data collection and analysis was not blinded. Images recorded for data analysis are indicated in the figure legends and correspond to the number of mice evaluated in each experiment.

## Histopathology and immunohistochemistry (IHC)

Tissue was collected from euthanized mice and stored in 10% neutral buffered formalin (NBF) for at least 24 hours. Samples were decalcified in TBD-2 (ThermoFisher) and dissected for paraffin processing and H&E staining per previously established protocols [10,11]. Slides were stained in Richard Allan Scientific Hematoxylin (Thermo Scientific) and Eosin-Y (Thermo Scientific) with the Leica Autostainer (Leica Biosystems). IHC staining was performed using rat anti-mouse F4/80 (1:200; AbD Serotec), rabbit anti-mouse MPO (1:400; Agilent), or rabbit anti-mouse IL-1β (1:200; Abcam) primary antibodies with the Intellipath Autostainer Immunostaining instrument. Slides were scanned using the Aperio ScanScope XT eSlide capture device (Aperio), and analyzed by Aperio ImageScope digital analysis software (v9.1) as detailed formerly [10,11]. Briefly, to quantify the extent of positive staining and lymphocyte localization, Aperio's positive pixel count algorithm was run using calibrated hue, saturation, and intensity values. Ten measurements of identical total surface area were quantitated to determine a mean positive pixel intensity value from each sample. All digital analysis was confirmed by histopathological evaluation by a board certified veterinary pathologist through the Comparative Pathology & Mouse Phenotyping Shared Resource at The Ohio State University using the 10x objective of a bright-field light microscope.

## Immunofluorescence

Paraffin sections were de-paraffinized using a xylene substitute (A5597, Sigma Life Science) and rehydrated with increasingly diluted ethanol. Heat-induced epitope retrieval was performed using sodium citrate buffer (10mM Sodium citrate, pH 6.0, Abcam) at sub-boiling temperatures for 10 minutes and cooled at room temperature for 30 minutes. The slides were incubated with rabbit anti-mouse IL-1β antibody (1:200, Abcam) in SignalStain antibody diluent (Cell Signaling Technology) at 4°C overnight. Goat anti-rabbit IgG A594 secondary antibody (1:100, Abcam) was applied for 1 hr at room temperature. Slides were subsequently stained with rat anti-mouse F4/80 antibody A488 (1:1000; ThermoFisher) and rabbit anti-mouse MPO A350 (1:500; Bioss) overnight at 4°C. ProLongTM Gold Antifade Mountant (ThermoFisher) was applied and slides were analyzed on the EVOS microscope (Thermo Scientific).

## Flow cytometry

Peripheral blood was isolated from mice and RBCs were lysed with ammonium chloride solution (Stemcell Technologies). Cells were subsequently plated and blocked with anti-mouse FcR antibody (αCD16/CD32, BioLegend) in FACS buffer (PBS with 2% BSA and 1mM EDTA). Cells were then surface-stained with CD11b-BV421 (clone M1/70, BioLegend), F4/80-APC (clone BM8, eBioscience), Ly6G-FITC (clone 1A8, Miltenyi Biotec), CD115-PerCP-Cy5.5 (clone AFS98, Biolegend), and TLR2-PE-Vio770 (clone REA109,

Miltenyi Biotec) according to manufacturer's recommendations. Data were collected on the BD LSRII Flow Cytometer (BD Biosciences) platform and exported for analysis via FlowJo (v.9.0; Treestar, Inc, Ashland, OR).

## Immunoblotting

Peripheral blood was collected via cardiac puncture and cells were stimulated *in vitro* following RBC lysis by LPS (Sigma, 100ng/mL) and MSU (Invivogen, 200ug/mL) for 6 hr prior to collection. Monocytes or neutrophils were isolated using immunomagnetic bead kits (STEMCELL Technologies) following manufacturer's protocol. Cells from multiple mice (n = 5) were pooled for each exercise condition, washed with PBS, and lysed in TN1 lysis buffer for protein analysis. Equal amounts of total protein were separated by SDS-PAGE gel electrophoresis, transferred to a PVDF membrane, and blocked. Membranes were then probed according to manufacturer's recommendations. Quantitation of band density and conversion to optical density values was performed using NIH ImageJ v1.53a software; values expressed relative to beta-actin normalization.

## ELISA

Serum was isolated and synovial aspirates were collected by rinsing joint spaces with PBS. ELISA was performed on diluted samples using the mouse IL-1β and CXCL1kits (eBioscience) according to manufacturer's protocol. Absorbance values were measured by the Dynex MRX-TC Revelation microplate reader/colorimeter (Dynex Technologies).

## Gout patient data

During scheduled appointments, gout patients not experiencing a flare at the time of visit were recruited from our clinics and consented to participate in the study (n = 30). International physical activity questionnaires (IPAQ) were completed to assess current levels of daily/weekly physical activity. Clinical data was collected during the patient visit, including BMI, age, years since diagnosis, attacks per year, perceived pain at the time of visit and in the past 4 weeks, and CRP levels. All patients signed written consent forms and were consented under Study ID: Protocol Pro20170000759 approved by IRB: New Brunswick Health Sciences through Rutgers Robert Wood Johnson Medical School. Participants were between the ages of 18 and 89 with a diagnosis of gout; exclusion criteria included inability to both read and speak English, significant mental impairment, psychiatric disorder that would limit ability to give informed consent or that might cause additional risks, a diagnosis of other autoimmune disease(s), or if current medications included steroids or other anti-inflammatory drugs. No minors were included in the study. Patients were recruited from July 2017 through July 2019. From the date of consent, information is accessible for use in study-related activities for 6 years.

## Statistical analysis

Statistical significance was determined in mouse studies using either unpaired t-tests (two-tail, equal SD) or ANOVA with multiple comparisons as described. In the human cohort, results were analyzed with individual unpaired t-tests with Welch's correction and one-way ANOVA without Gaussian distribution and multiple comparisons to analyze the mean rank of each data set with that of every other to obtain p-values. Statistical significance was determined to be $p < 0.05$. Analysis was completed using GraphPad Prism (v.6.0).

# Results

## Establishment of an acute gout model

MSU crystals were injected into the ankle joint of BALB/C-Tg(NFκB-RE-luc)-Xen male mice to monitor inflammation both broadly and at the site of MSU crystal injection [9,12]. Male mice were selectively used in this study to investigate inflammatory pathogenesis and disease pathology in order to better reflect the human patient population. Epidemiological data shows that the male sex-bias in the disease reaches disparities of 10:1 when considering pre-menopausal cohorts [13]. Phase contrast and polarizing light microscopy detected MSU crystals and cells were observed actively engulfing crystals in synovial aspirates 16 hrs post-injection (S1 Fig). Mice receiving MSU crystal injections exhibited significantly increased ($p < 0.01$) ankle swelling ($0.386 \pm 0.107$mm) compared to the PBS-injected controls ($0.067 \pm 0.153$mm) (Fig 1A). Using the *in vivo* Imaging System (IVIS), localized NF-κB activity in the ankle synovial region was compared between PBS and MSU crystal-injected mice. Bioluminescent imaging revealed a 69% ($p < 0.05$) increase in NF-κB activity in the MSU crystal group (Fig 1B). Furthermore, whole-mouse imaging confirmed that MSU-induced inflammation is indeed localized to the injected ankles (Fig 1C). Histological examination by H&E staining revealed a visibly robust inflammatory infiltrate in the synovial space of the injected ankle in the MSU crystal group compared to the PBS group (Fig 1D and 1E). Since it has been reported that macrophages and neutrophils are among the primary effectors of inflammation in chronic gouty arthritis [14,15],

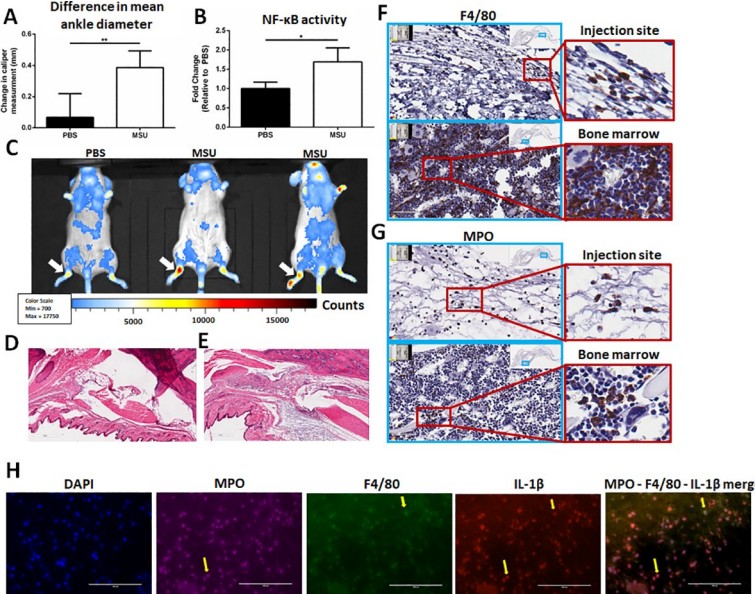

**Fig 1. Establishment of an acute gout model in male mice with a bioluminescent reporter associated with NF-κB activity. (A)** Change in mean ankle diameter 15 hrs post-PBS/MSU injection. Values reported as changes in caliper measurement (final–initial) average ± SD. (n = 3; n = 7, respectively) **(B)** NF-κB activity was compared in injected feet by *in vivo* imaging (IVIS) between PBS and MSU treatments. Values reported as fold change (FC ± SD) relative to PBS (n = 3 PBS; n = 5 MSU). Analysis by two-tailed, nonparametric, unpaired Mann-Whitney t-test. *$p < 0.05$, **$p < 0.01$. **(C)** Whole-mouse comparison of bioluminescent signals between PBS (n = 3) and MSU (n = 5) crystal injected mice (white arrows indicate the site of injection). **(D-E)** Hematoxylin&Eosin (H&E) stains at 100x magnification in Aperioimage software of **(D)** PBS-injected and **(E)** MSU-injected ankle joint spaces. **(F)** F4/80 immunohistochemistry (IHC) of joint space (top) and bone marrow (bottom). **(G)** MPO IHC of joint space (top) and bone marrow (bottom). Images on left are 100x magnification and images on right are 400x magnification. Scanned using in Aperio image software. Bone marrow images represent positive controls for IHC staining. **(H)** Immunofluorescent staining of DAPI, MPO, F4/80, and IL-1β within the synovium of MSU-injected mice. Yellow arrows highlight co-localization of IL-1β expression for macrophages and neutrophils.

IHC for the macrophage marker F4/80 and the granulocyte marker myeloperoxidase (MPO) were measured (Fig 1F and 1G). Staining revealed the presence of both F4/80[+] macrophages and MPO[+] neutrophils in the synovium of MSU-crystal injected mice. To confirm that these infiltrating macrophages and neutrophils recapitulate predicted pathology by expressing IL-1β [16], immunofluorescent staining was performed on ankle synovial tissue sections to demonstrate co-localization of IL-1β expression with F4/80[+] macrophages and MPO[+] cells (Fig 1H). Collectively, these results demonstrate that intra-articular MSU crystal injections can induce a localized inflammatory response characterized by increased edema, NF-κB activity, and tissue infiltration by IL-1β-expressing macrophages and neutrophils.

## Exercise intensity correlates with severity of inflammation

To determine treadmill speeds correlating with different exercise intensities, a metabolic treadmill and respirometer were used to measure maximum oxygen consumption ($VO_2$ max) in male and female BALB/c mice. Since no differences were observed between sexes in analysis, the data were reported collectively. Low, medium, and high-intensity exercise were defined as 35% $VO_2$ max, 55% $VO_2$ max, and 75% $VO_2$ max, respectively [17]. A polynomial curve of $VO_2$ max values at increasing treadmills speeds was used to interpolate exercise intensity levels. Our results showed that the speeds for 35%, 55%, and 75% $VO_2$ max in this mouse strain were 8 m/min, 11 m/min, and 15 m/min, respectively (S2 Fig). Additionally, although murine exercise studies have shown that physical activity can induce a positive effect on emotional behavior by promoting neurotrophic factor expression [18]. no differences in emotional behavior were observed in the mice examined in this study.

Caliper measurements of ankle diameter obtained before and after MSU crystal injection in male mice revealed a significant increase in the non-exercised control mice (0.82 ± 0.27mm). In comparison, mice that exercised before MSU injection exhibited less inflammation compared to the non-exercised control after 15 hrs, but to varying degrees. The low (0.22 ± 0.20mm; p<0.0001) and moderate-intensity (0.13 ± 0.21mm; p<0.0001) exercise groups displayed comparable levels of ankle swelling, which were both significantly less than the non-exercised control and high-intensity groups (Fig 2A).

Comparisons of relative NF-κB activity using IVIS revealed a similar trend (Fig 2B). Relative to the non-exercised controls, male NF-κB-luc reporter mice in the low and moderate-intensity groups had 54 ± 15% (p < 0.0001) and 60 ± 15% (p < 0.0001) reduced levels of NF-κB activity, respectively. Furthermore, there was less than a 10% change in NF-κB activity between the non-exercised control and the high-intensity exercise group. Since physical activity as an intervention itself has been shown to correlate with decreased inflammation, male and female mice were exercised at low, moderate, and high intensities; bioluminescent imaging was performed both pre and post-exercise to measure NF-κB activity for comparison to the non-exercised control group. Measurements of systemic NF-κB activity in mice showed no significant differences across groups (S3 Fig), which demonstrates that these exercise regimens do not significantly influence endogenous levels of inflammation. Collectively, these results suggest that exercise intensity can greatly affect the amount of inflammation and NF-κB activity in the synovium following ankle injection with MSU crystals.

## Low to moderate-intensity exercise decreases macrophage and neutrophil infiltration *in situ*

To determine if exercise intensity can affect the degree of macrophage/neutrophil infiltration, we examined the macrophage marker F4/80 and the granulocyte marker MPO on paraffin-embedded sections of the ankle synovial space (Fig 3). F4/80[+] cells are decreased in the

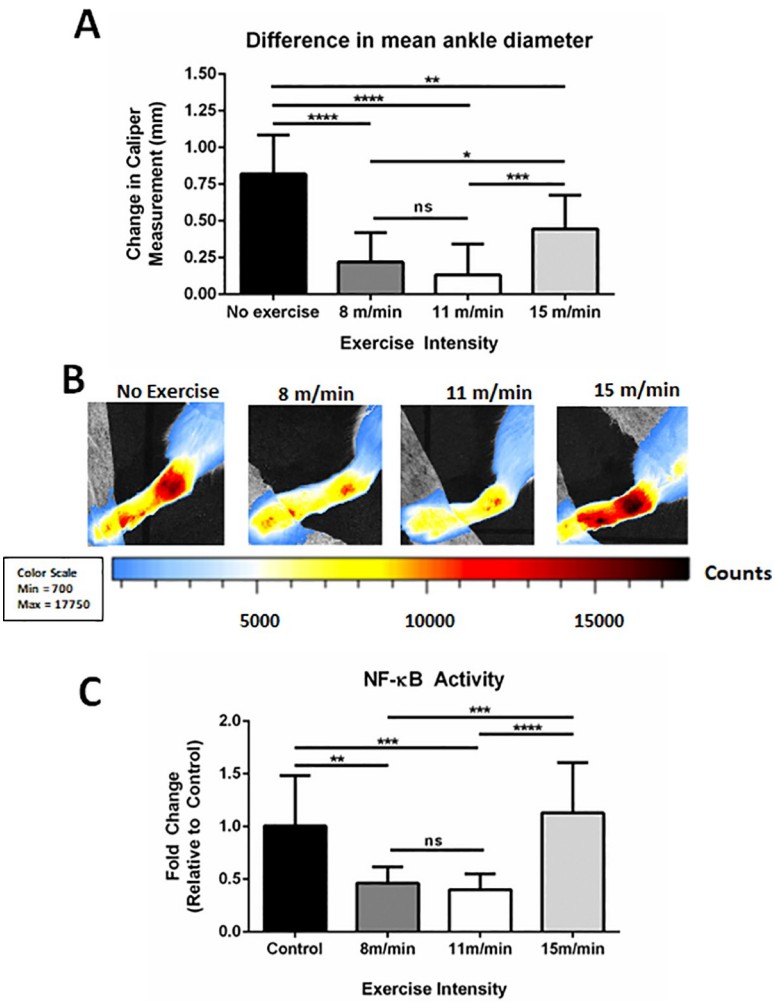

**Fig 2. Mean difference in ankle diameter and NF-κB activity in MSU-injected mice following exercise regimens of varying intensity. (A)** Change in mean ankle diameter (swelling) between non-exercised, 8 m/min, 11 m/min, and 15 m/min groups. Values reported as changes in caliper measurement (final–initial) average ± SD (n = 10, 10, 22, 11, respectively). **(B)** Representative *in vivo* imaging system (IVIS) images of MSU injected ankles from non-exercised (n = 10), 8 m/min (n = 10), 11 m/min (n = 22), and 15 m/min (n = 10) groups. Scale of luminescent counts below. **(C)** Normalized IVIS intensity values as fold changes relative to non-exercised controls (FC ± SD). (n = 10, 10, 22, 11 respectively). Analysis by ANOVA and followed by two-tailed, nonparametric, unpaired Mann-Whitney t-tests. *p<0.05, **p<0.01, ***p<0.001, ****p<0.0001, ns = not significant.

synovium of low and moderate-intensity exercise groups compared to the non-exercised controls and the high-intensity exercise group (Fig 3A). To quantify these differences, the slides were scanned and digitized for histopathological analysis using Aperio IHC analysis software [10]. F4/80 staining within the intra-articular space shows that there is 68 ± 33% less F4/80 staining in the low-intensity exercise group (p<0.01) and 93 ± 5% less F4/80 staining in the moderate-intensity exercise group (p<0.001) relative to the non-exercised controls and the high-intensity exercise group. (Fig 3B). Further histopathological analysis of the joint space for the granulocyte marker MPO revealed decreased MPO staining in the low-intensity (69 ± 21%, p<0.01) and moderate-intensity (83 ± 30%, p<0.001) groups relative to the high-intensity exercise group and the non-exercised controls (Fig 3C and 3D). In parallel with the F4/80 staining, there was no significant difference in MPO staining between the high-intensity

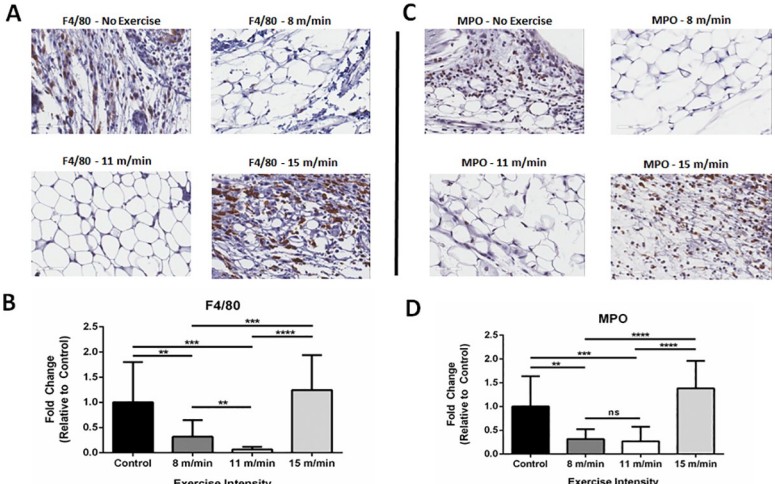

**Fig 3. Immunohistochemistry for F4/80 and MPO in MSU-injected synovium. (A)** Expression of F4/80 by immunohistochemical (IHC) staining on non-exercised control, 8 m/min, 11 m/min, and 15 m/min exercise conditions. Image magnification at 400x and pictures captured from Aperio Image software. **(B)** Analysis of F4/80+ pixel intensity of 10 randomized areas in joint space, as expressed by fold change (FC ± SD) relative to non-exercised controls (n = 3 mice/group). **(C)** Expression of myeloperoxidase (MPO) by IHC on non-exercised control, 8 m/min, 11 m/min, and 15 m/min exercise conditions. Image magnification at 400x with Aperio Image software photomicrographs. **(D)** Analysis of MPO+ pixel intensity of 10 randomized areas in joint space as expressed by fold change (FC ± SD) relative to non-exercised controls (n = 3 mice/group). Analysis by ANOVA and followed by two-tailed, nonparametric, unpaired Mann-Whitney t-tests. *$p < 0.05$, **$p < 0.01$, ***$p < 0.001$, ****$p < 0.0001$, ns = not significant.

exercise group and the non-exercised controls. These results suggest that exercise intensity plays a significant role in modulating the ability of macrophages and neutrophils to infiltrate the synovium in response to inflammatory stimuli, such as MSU crystals.

## IL-1β expression is decreased locally and inflammasome activation is downregulated systemically following low/moderate-intensity exercise

IL-1β is a tightly regulated inflammatory cytokine, which has been previously shown to be a primary mediator of MSU crystal-induced inflammation and inflammasome activation [19–21]. To determine the extent exercise intensity can affect IL-1β, systemic and localized IL-1β expression levels were examined. While IL-1β expression levels in serum of MSU-injected NF-κB-luc reporter mice or wild-type control Balb/C mice did not differ significantly from synovial aspirates of PBS-injected mice, localized IL-1β expression in the synovium of NF-κB-luc mice was significantly elevated after MSU injection (Fig 4A). Furthermore, an examination of cleaved/secreted IL-1β by IHC at the site of MSU crystal injection revealed markedly reduced IL-1β staining in the synovial space of low/moderate-intensity exercise (Fig 4B). Quantification using Aperio IHC analysis revealed a 68.5% (p<0.0001) and 86.1% (p<0.0001) reduction in IL-1β staining present locally in the synovium of low or moderate-intensity exercise groups, respectively (Fig 4C). There was no significant difference in IL-1β staining between the high-intensity exercise group and the non-exercised controls. Collectively, these results show that MSU-induced IL-1β expression is suppressed locally with low and moderate exercise.

In a previous murine model of gouty inflammation, neutrophil infiltration and IL-1β production at the site of MSU crystal injection were shown to be suppressed in mice lacking various components of the NLRP3 inflammasome [19]. Given the reduced presence of IL-1β, monocytes, and neutrophils at the site of MSU crystal injection, inflammasome activity was examined

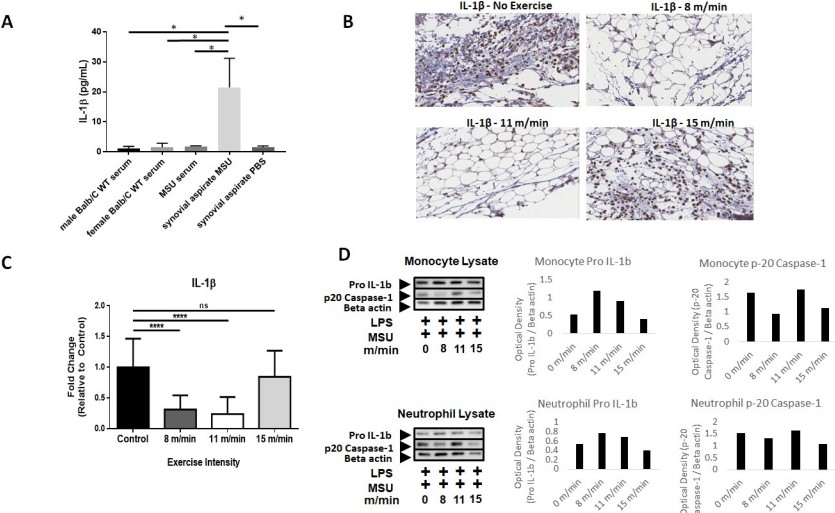

**Fig 4. IL-1β expression in MSU-injected mice and *in vitro* analysis of inflammasome activation in monocytes and neutrophils. (A)** ELISA analysis of IL-1β expression in serum or synovial aspirates of Balb/c mice. **(B)** Expression of IL-1β by immunohistochemical (IHC) staining on non-exercised control, 8 m/min, 11 m/min, and 15 m/min exercise conditions. Image magnification at 400x and pictures captured using Aperio Image software. **(C)** Analysis of F4/80+ pixel intensity of 10 randomized areas in joint space as expressed by fold change (FC ± SD) relative to non-exercised controls (n = 3 mice/group). **(D)** Following exercise regimens at low, medium, and high intensities, neutrophils and monocytes were isolated from whole blood by immunomagnetic sorting and stimulated *in vitro* by LPS priming followed by MSU to induce inflammasome activation. Western blots were performed on pooled cell lysates (n = 5 per condition) to measure pro-IL-1β and p20 caspase-1 expression. Beta-actin was used as a loading control. Optical density values expressed relative to beta-actin normalization. Analysis by ANOVA and followed by two-tailed, nonparametric, unpaired Mann-Whitney t-tests. *p<0.05, **p<0.01, ***p<0.001, ****p<0.0001, ns = not significant.

in peripheral monocytes and neutrophils to determine if exercise intensity can affect inflammasome activation upstream of IL-1β production and cellular infiltration. Mice were exercised at low, medium, and high intensities daily for 2 weeks. Whole blood samples from each condition were isolated by positive selection with immunomagnetic sorting and macrophages or neutrophils were treated with LPS and MSU *in vitro* to stimulate inflammasome activation. Cell lysates were analyzed by Western blotting for changes in pro-IL-1β production and activation of p20 caspase-1 following isolation with immunomagnetic beads. With LPS and MSU stimulation, both neutrophils and monocytes had elevated levels of pro-IL-1β following low/moderate intensity exercise relative to high intensity and controls without exercise (Fig 4D). Additionally, previous studies have shown that inflammasome activation is directly correlated with the level of p20 active caspase-1 released from cells into the surrounding extracellular space [19]. Thus, higher levels of p20 caspase-1 in the cellular lysate indicates that p20 caspase-1 is not secreted, which suggests decreased inflammasome activation. Mice that exercised at 11 m/min had elevated p20 caspase-1 remaining in the neutrophil and monocyte lysates compared to the other experimental groups (Fig 4D). These data suggest that both low and moderate exercise regimens can result in physiologically dysregulated inflammasome activation when exposed to catalysts to this response *in vitro*, which may be associated with reduced levels of inflammatory recruitment of neutrophils and monocytes locally in MSU crystal-induced arthritis.

## Low and moderate-intensity exercise decreases systemic levels of CXCL1 and TLR2 expression on peripheral neutrophils

Previous studies in humans have shown that moderate physical activity (~65% VO$_2$ max) can decrease the surface expression of TLRs on peripheral monocytes and neutrophils [21].

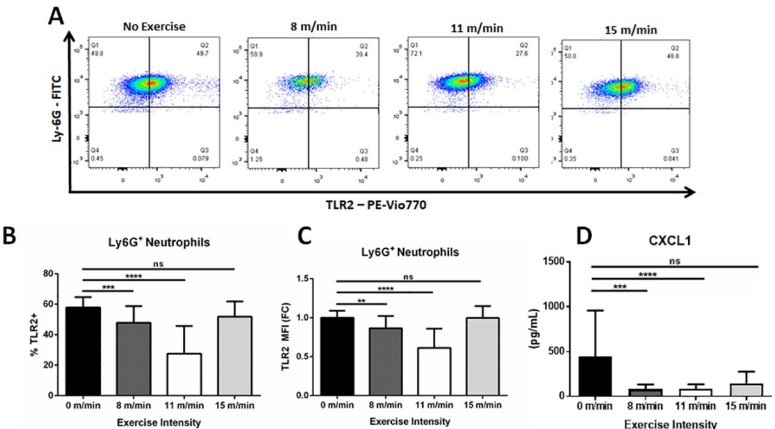

**Fig 5. Analysis of and CXCL1 serum levels and TLR2 expression on peripheral neutrophils. (A)** Flow cytometry of TLR2 expression on Ly6G+ / CD115- neutrophils from the periphery of non-exercised control, 8 m/min, 11 m/min, and 15 m/min groups. Images are representative of multiple samples. **(B)** Analysis of percent TLR2+ neutrophils and **(C)** TLR2 MFI from non-exercised, 8 m/min, and 11 m/min conditions. (n = 10, 15, 15, and 10 samples run in duplicate, respectively). **(D)** ELISA analysis measuring CXCL1 expression in serum. Statistical analysis by ANOVA and followed by two-tailed, nonparametric, unpaired Mann-Whitney t-tests. Error bars ±SD. $^*p<0.05$, $^{**}p<0.01$, $^{***}p<0.001$, $^{****}p<0.0001$, ns = not significant.

Furthermore, TLR2 signaling has been highly implicated in mediating gout pathogenesis [22–24]. Using flow cytometry, we quantified TLR2 expression on peripheral monocytes and neutrophils following exercise and MSU crystal-induced gout (Fig 5A). A finalized gating strategy for TLR2 detection was determined by using compensation control analysis with both unstained and single stained cells (S4 Fig). A significant decrease in TLR2 expression on neutrophils was observed in both low-intensity (47.81 ± 10.98%; p<0.001) and moderate-intensity (27.59 ± 18.15%; p<0.0001) exercise groups compared to the non-exercised controls (57.91 ± 6.79%) (Fig 5B). Similarly, the median-fluorescent intensity (MFI) of TLR2 on neutrophils was significantly decreased in the low-intensity (13.41 ± 15.56%; p<0.001) and moderate-intensity (38.69 ± 24.73%; p<0.0001) groups compared to the non-exercised controls (Fig 5C). There was less than 1% difference in MFI between the high-intensity exercise mice and the non-exercised controls. These results confirm that TLR2 expression on neutrophils is significantly affected by exercise intensity and provides a plausible mechanism for the observed decreases in inflammation and NF-κB activity *in vivo* in this model.

Our histopathological results demonstrate a decrease in neutrophil infiltrates at the site of MSU crystal injection with low/moderate physical activity. CXCL1 is a neutrophil chemoattractant that recruits neutrophils and thereby helps facilitate inflammatory responses [25]. Our results show that whole-blood expression of CXCL1was significantly suppressed in the low-intensity (80.18 ± 13.61 pg/mL; p<0.01) and moderate-intensity (84.65 ± 11.95 pg/mL p<0.01) groups compared to the non-exercised controls (440.9 ± 110.3 pg/mL), respectively (Fig 5D). No significant differences were observed between high-intensity exercise mice and the non-exercised controls. These data demonstrate that low/moderate intensity exercise results in a systemic suppression of CXCL1 expression with the stimulation of MSU crystal-induced arthritis, which can lead to the inhibition of neutrophil recruitment locally to ankle joint synovial space.

## Increased physical activity in human gout patients correlates with decreased markers of inflammation and better patient prognosis

While exercise has recently been shown to be efficacious in reducing pathogenic inflammatory burdens in several autoimmune diseases, the impact in patients with gout has not been studied

extensively. To examine the effects of physical activity in a gout patient population, physical activity questionnaires and clinical data were analyzed from a cohort of patients with a gout diagnosis (n = 30). The patient population consisted of 25 males and 5 females; however, all females were post-menopausal. While the male predilection of gout is 10:1 prior to menopause, this is reduced significantly in post-menopausal cohorts presumably due to the uricosurics effect of female sex hormones [13]. Furthermore, since individual analysis showed no significant correlation to sex in any metric measured, the data from this cohort was reported collectively. At the scheduled visit, gout patients not experiencing a flare at the time of visit were recruited from our clinics; IPAQ surveys were completed to assess current levels of daily/weekly physical activity and clinical data was obtained, including BMI, age, years since diagnosis, attacks per year, perceived pain at the time of visit and in the past 4 weeks, and CRP levels. The IPAQ survey, which has been previously used in studies of rheumatologic diseases [26], separated the gout patients into physically active (n = 16) and physically inactive cohorts (n = 14) (p<0.001; Fig 6A). While average age, BMI, or years since diagnosis did not significantly differ between cohorts (Fig 6B–6D), physically active gout patients had over 12-fold fewer gout flares per year (p<0.01; Fig 6E), 10-fold less CRP (p<0.01; Fig 6F), a 4.6-fold decrease in perceived pain at the time of visit (p<0.01; Fig 6G), and a 2.8-fold decrease in perceived pain over the past 4-week period (p<0.05; Fig 6H). These data indicate that increased physical activity is indeed beneficial in patients with gout.

## Discussion

Our results show that regimens of low-to-moderate intensity exercise can significantly reduce the inflammation observed in an acute model of murine gout. Mice receiving low-intensity

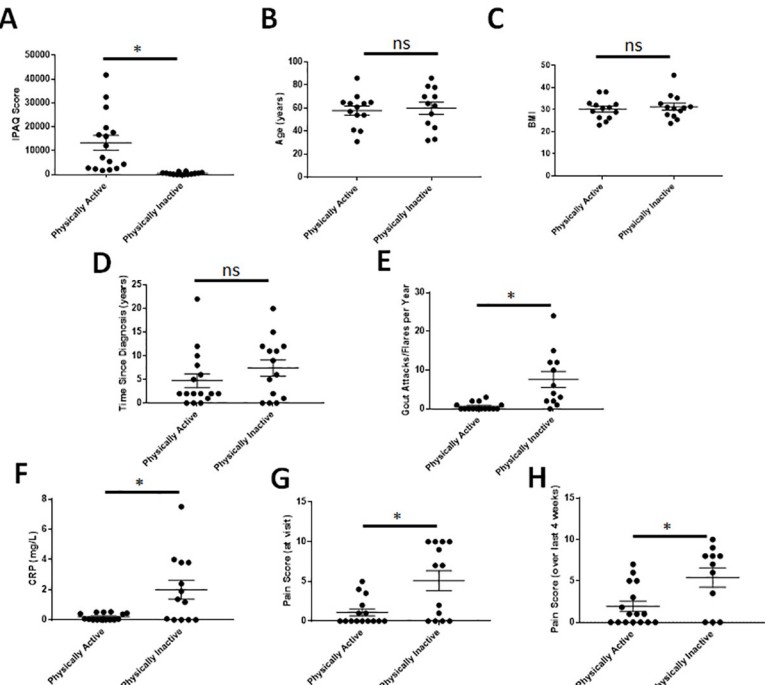

**Fig 6. Questionnaires and clinical data collected from gout patients.** Physically active = 16 (IPAQ>1400); physically inactive = 14 (IPAQ<1400). Data analyzed with individual unpaired t-tests with Welch's correction and one-way ANOVA without Gaussian distribution and multiple comparisons to analyze the mean rank of each data set with that of every other to obtain p-values. p≤0.05 considered statistically significant. *p<0.05, **p<0.01, ***p<0.001, ****p<0.0001, ns = not significant.

and moderate-intensity exercise had measurably less swelling and NF-κB activity than the non-exercised controls. Furthermore, we showed that low-to-moderate exercise significantly decreases infiltration of F4/80[+] macrophages and MPO[+] neutrophils into the synovium of MSU crystal-injected mice. As a possible mechanism explaining these results, we show that mice receiving low-intensity and moderate-intensity exercise have significantly fewer TLR2[+] neutrophils and reduced systemic expression of the neutrophil chemoattractant CXCL1 relative to non-exercised controls.

Research studies specifically examining the effects of exercise in a canine model of gouty arthritis date back almost half a century, were performed only once, and contained very small sample sizes [27,28]. Agudelo and Dorwart reported increased articular inflammation and tissue infiltration in the exercised condition versus the non-exercised condition [28,29]. This is likely attributable to experimental differences in MSU crystal administration: Agudelo and Dorwart injected MSU crystals prior to initiating joint movement whereas mice in this study were physically conditioned to pre-determined exercise intensities, exercised regularly for two weeks, and then administered MSU crystals. Additionally, these studies used passively controlled movement of the effected joint in an otherwise resting animal while our study evaluated the effects of physical activity with active ambulatory exercise routines.

MSU crystals induce cellular stress resulting in NLRP3 inflammasome activation and IL-1β secretion [20,30]. In addition, *in vitro* studies with human and murine macrophages have shown that free fatty acids can signal through TLR2 and prime the MSU-induced immune response synergistically leading to ASC-caspase-1 driven IL-1β release [20]. Our results show decreased TLR2 expression on the surface of neutrophils, which suggests that exercise downregulates this priming element of inflammasome activation. Moreover, we demonstrated *in vitro* that low/moderate exercise leads to an upregulation of p20 caspase-1 and/or pro-IL1β expression intracellularly, suggesting dysregulated inflammasome signaling. Furthermore, previous studies have shown that chronic, low-level exposure to LPS priming of cells *in vitro* leads to a suppression of inflammasome activation and IL-1β expression [31]. Since physical activity is associated with an acute pro-inflammatory response [32], this can potentially function as a daily chronic exposure to inflammation physiologically and result in immune tolerance to subsequent challenges with MSU crystals. Collectively, these data suggest that exercise-induced downregulation of TLR2 expression and inflammasome priming may result in a tolerogenic effect and reduced inflammatory responses. Thus, we propose that the reduced inflammation observed with MSU crystal injection is not an anti-inflammatory response, but rather an immunomodulated and regulated immune response resulting from immune tolerance derived from chronic exposures to small bouts of inflammation following exercise. Using this hypothesis, it can be speculated that exercise can immuno-optimize physiologically by preventing hyperactive immune responses leading to inflammation and clinical symptoms following antigenic challenge, which would correlate with the wealth of epidemiological data concluding that people with higher levels of physical activity have a significantly decreased frequency of clinical illness.

While the clinical practice guidelines for gout intervention released by The European League Against Rheumatism (EULAR; 2016) suggest that regular physical activity may help combat the mortality associated with chronic hyperuricemia, no insight is provided into intensity, duration, or type of exercise. Current ACR and EULAR recommendations for the treatment of gout have only recently started to recommend exercise for the treatment of gout; however, no recommendations have been made regarding the type or intensity of exercise [6,33]. Considering the improved gout patient clinical course and CRP levels observed in our human cohort and the increasing prevalence and incidence of gout, this further highlights the importance of incorporating low or moderate-intensity exercise as an adjunct treatment option [34].

Whether exercise can affect the frequency of gout flares in a mouse model has yet to be determined and will require the establishment and characterization of a chronic gout model, which more accurately represents disease occurrence in humans.

In summary, this study reveals that low-intensity and moderate-intensity exercise can prevent the inflammatory response in acute models of murine gout. Low and moderate-intensity exercise can reduce tissue swelling, NF-κB activity, tissue infiltration by macrophages and neutrophils, TLR2 expression on peripheral neutrophils, and systemic expression of CXCL1. Although this is not the first study to examine the relationship between exercise and gout, our work helps define the therapeutic potential of low/moderate-intensity aerobic exercise in the management of gout and other inflammatory diseases. In subsequent experiments to the work outlined here, we also examined the effects of exercise during and after the induction of gout in preliminary studies to determine if exercise can therapeutically inhibit the pathology associated with flares. NF-κB-RE-luc mice were exercised at the optimal regimen defined herein (11 m/min) and MSU crystals were injected intra-articularly in several rounds with a rest/recovery period in-between. The results indicated complete resolution of localized NF-κB activity in mice given repeated MSU injections without measures to raise serum uric acid levels (S5 Fig); thus, a chronic model of gout could not be established, which precluded further characterization.

Elucidating the role exercise plays in health and disease is fundamental to our understanding of human physiology and inflammatory disease pathogenesis. As our knowledge of the benefits of exercise increases at the molecular level, so too will our ability to incorporate its effectiveness as a tool for improving health outcomes in the clinical setting. Considered collectively and in a translational context, we feel that the results of our study show that exercise would be helpful to a patient with gout that is in a recovery period between flares to hopefully prevent or limit future ones, as a preventative measure. While rest and decreased movement/weight may still be recommended to a patient with a flare presenting with a red, painful, and swollen foot, we envision a standardized exercise regimen being prescribed during times of clinical inactivity to help the severity and frequency of future occurrences. Future translational studies examining the effects low and moderate-intensity exercise in patients with gout could facilitate new discoveries in our understanding and treatment of this disease.

## Supporting information

**S1 Fig. Confirmation of MSU crystals in the ankle joints of MSU injected mice (top row).** MSU crystals were visualized by polarizing light and phase contrast microscopy following resuspension in PBS, diluted 1:50, or PBS without MSU. Mice were given intra-articular injections of PBS or MSU and assessed 16 hrs later. **(middle row)** Pictures taken of feet/ankles show increased swelling with MSU injection relative to contralateral feet/ankles or PBS controls (white arrows). **(bottom row)** Synovial aspirates of PBS-injected mice show no detectable cells or MSU crystals. Conversely, MSU injection produced detectable MSU crystals and cells were observed actively engulfing crystals (white arrow).
(TIF)

**S2 Fig. Determination of exercise intensity by respirometry.** Male and female Balb/c mice (n = 14) were exercised in five minutes intervals at increasing speeds with a 5% incline. The VO$_2$ consumption was recorded every 30 seconds. Values were plotted and a polynomial curve was applied to graph and interpolate speeds correlating with 35% (low-intensity), 55% (medium-intensity), and 75% (high-intensity) VO$_2$ max. Speeds were determined to be 8 m/min, 11 m/min, and 15 m/min, respectively.
(TIF)

**S3 Fig. Exercise alone does not significantly impact NF-κB activity.** Male and female Balb/c mice (n = 14) were exercised at 8 m/min, 11 m/min, and 15 m/min daily for 2 weeks. Systemic NF-κB activity was measured by IVIS before beginning the exercise regimen and 24 hrs after the completion of the final session. Analysis by ANOVA and followed by two-tailed, nonparametric, unpaired Mann-Whitney t-tests produced no statistically different results.
(TIF)

**S4 Fig. Flow cytometry gating strategy for the detection of TLR2.** Gating strategy of successive flow cytometry gates to detect TLR2 positive expression. The total cell population was first gated on forward and side scatter. Subsequently, neutrophils were gated by Ly6G+ / CD11b + detection. TLR2 expression was then determined by PE-Vio770 fluorochrome detection relative to the signals observed in unstained and single stained cells.
(TIF)

**S5 Fig. Repeated MSU injections do not contribute to enhanced NF-κB activation.** Bioluminescent imaging of NF-κB-RE-luc mouse feet measured and quantitated via IVIS. Repeated intra-articular injections of MSU crystals (20 μg) were made where indicated. Analysis of localized NF-κB activity suggests complete resolution prior to subsequent challenges.
(TIF)

**S1 Raw data.**
(ZIP)

**S1 File.**
(DOC)

## Acknowledgments

We would like to thank the Comparative Pathology & Mouse Phenotyping Shared Resource staff, Department of Veterinary Biosciences staff, and the Comprehensive Cancer Center staff at The Ohio State University.

## Author Contributions

**Conceptualization:** Kyle Jablonski, Nicholas A. Young, Anuradha Kalyanasundaram, Emmy Schwarz, Paul Consiglio, Anna Bratasz, Anasuya Sarkar, Amal O. Amer, Naomi Schlesinger.

**Data curation:** Kyle Jablonski, Nicholas A. Young, Caitlin Henry, Kyle Caution, Anuradha Kalyanasundaram, Peter Harb, Emmy Schwarz, Paul Consiglio, Anna Bratasz, Amal O. Amer.

**Formal analysis:** Kyle Jablonski, Nicholas A. Young, Anuradha Kalyanasundaram, Paul Consiglio, Anna Bratasz, Anasuya Sarkar, Amal O. Amer, Wael N. Jarjour.

**Funding acquisition:** Nicholas A. Young, Naomi Schlesinger.

**Investigation:** Kyle Jablonski, Nicholas A. Young, Anuradha Kalyanasundaram, Paul Consiglio, Anna Bratasz, Wael N. Jarjour.

**Methodology:** Kyle Jablonski, Nicholas A. Young, Caitlin Henry, Kyle Caution, Anuradha Kalyanasundaram, Paul Consiglio, Chris M. Cirimotich, Anna Bratasz, Anasuya Sarkar, Amal O. Amer, Wael N. Jarjour.

**Project administration:** Kyle Jablonski, Nicholas A. Young, Ifeoma Okafor, Peter Harb, Emmy Schwarz.

**Resources:** Kyle Jablonski, Nicholas A. Young, Caitlin Henry, Ifeoma Okafor, Wael N. Jarjour.

**Supervision:** Nicholas A. Young.

**Visualization:** Nicholas A. Young, Emmy Schwarz, Chris M. Cirimotich, Anna Bratasz.

**Writing – original draft:** Kyle Jablonski, Nicholas A. Young, Caitlin Henry, Kyle Caution, Anuradha Kalyanasundaram, Ifeoma Okafor, Peter Harb, Emmy Schwarz, Paul Consiglio, Chris M. Cirimotich, Anna Bratasz, Anasuya Sarkar, Amal O. Amer.

**Writing – review & editing:** Kyle Jablonski, Nicholas A. Young, Caitlin Henry, Kyle Caution, Anuradha Kalyanasundaram, Ifeoma Okafor, Peter Harb, Emmy Schwarz, Paul Consiglio, Chris M. Cirimotich, Anna Bratasz, Anasuya Sarkar, Amal O. Amer, Wael N. Jarjour, Naomi Schlesinger.

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
