## [Decision Letter · Decision Letter 0]

8 Apr 2020

PONE-D-20-02077

Physical Activity Suppresses Acute Inflammation in a Gout Model by Downregulation of TLR2 on Circulating Neutrophils as well as Inhibition of Serum CXCL1 and is Associated with Decreased Pain and Inflammation in Gout Patients

PLOS ONE

Dear Dr. Young,

Thank you for submitting your manuscript to PLOS ONE. After careful consideration, we feel that it has merit but does not fully meet PLOS ONE’s publication criteria as it currently stands. Therefore, we invite you to submit a revised version of the manuscript that addresses the points raised during the review process.

We would appreciate receiving your revised manuscript by May 23 2020 11:59PM. To enhance the reproducibility of your results, we recommend that if applicable you deposit your laboratory protocols in protocols.io, where a protocol can be assigned its own identifier (DOI) such that it can be cited independently in the future. For instructions see: http://journals.plos.org/plosone/s/submission-guidelines#loc-laboratory-protocols

We look forward to receiving your revised manuscript.

Kind regards,

Fulvio D'Acquisto, PhD

Academic Editor

PLOS ONE

Journal Requirements:

2.Please provide additional details regarding participant consent. In the ethics statement in the Methods and online submission information, please ensure that you have specified whether consent was written or verbal/oral. If consent was verbal/oral, please specify: 1) whether the ethics committee approved the verbal/oral consent procedure, 2) why written consent could not be obtained, and 3) how verbal/oral consent was recorded. If your study included minors, please state whether you obtained consent from parents or guardians in these cases.

3. Please include in your Methods section the date ranges over which you recruited participants to this study."

4. To comply with PLOS ONE submissions requirements, please provide method(s) of animal sacrifice/euthanasia in the Methods section of your manuscript.'"

5. Please note that all PLOS journals ask authors to adhere to our policies for sharing of data and materials: https://journals.plos.org/plosone/s/data-availability. According to PLOS ONE’s Data Availability policy, we require that the minimal dataset underlying results reported in the submission must be made immediately and freely available at the time of publication. As such, please remove any instances of 'unpublished data' or 'data not shown' in your manuscript and replace these with either the relevant data (in the form of additional figures, tables or descriptive text, as appropriate), a citation to where the data can be found, or remove altogether any statements supported by data not presented in the manuscript.

6. We note that you have indicated that data from this study are available upon request. PLOS only allows data to be available upon request if there are legal or ethical restrictions on sharing data publicly. For more information on unacceptable data access restrictions, please see http://journals.plos.org/plosone/s/data-availability#loc-unacceptable-data-access-restrictions.

7. Thank you for stating the following financial disclosure:

"Both Nicholas Young and Naomi Schlesinger had research support initially derived from Ardea Biosciences (this support was subsequently supplemented via Ironwood Pharmaceuticals) examining the role of exercise in inflammation, which directly supported these research endeavors. We would like to extend our gratitude to the Clinical Research Center and the Research Match Program through the Center for Clinical and Translational Science (CCTS) at OSUWMC. The CCTS is supported by UL1TR001070 from the National Center for Advancing Translational Science. Comparative Pathology & Mouse Phenotyping Shared Resource at The Ohio State University is supported in part by grant P30 CA016058, National Cancer Institute."

8. Thank you for stating the following in the Financial Disclosure section:

"Both Nicholas Young and Naomi Schlesinger had research support initially derived from Ardea Biosciences (this support was subsequently supplemented via Ironwood Pharmaceuticals) examining the role of exercise in inflammation, which directly supported these research endeavors. We would like to extend our gratitude to the Clinical Research Center and the Research Match Program through the Center for Clinical and Translational Science (CCTS) at OSUWMC. The CCTS is supported by UL1TR001070 from the National Center for Advancing Translational Science. Comparative Pathology & Mouse Phenotyping Shared Resource at The Ohio State University is supported in part by grant P30 CA016058, National Cancer Institute."

We note that you received funding from a commercial source: 'Ironwood Pharmaceuticals'

9. Thank you for stating the following in the Competing Interests section:

"Naomi Schlesinger has grants from Pfizer and AMGEN; she is also on the advisory board and consulting for Novartis, Horizon Pharma, Selecta Biosciences, Olatec, and Mallinckrodt Pharmaceuticals. The remaining authors declare no conflict of interest."

10. PLOS ONE now requires that authors provide the original uncropped and unadjusted images underlying all blot or gel results reported in a submission’s figures or Supporting Information files. This policy and the journal’s other requirements for blot/gel reporting and figure preparation are described in detail at https://journals.plos.org/plosone/s/figures#loc-blot-and-gel-reporting-requirements and https://journals.plos.org/plosone/s/figures#loc-preparing-figures-from-image-files. When you submit your revised manuscript, please ensure that your figures adhere fully to these guidelines and provide the original underlying images for all blot or gel data reported in your submission. See the following link for instructions on providing the original image data: https://journals.plos.org/plosone/s/figures#loc-original-images-for-blots-and-gels.

Reviewers' comments:

Reviewer's Responses to Questions

**Comments to the Author**

1. Is the manuscript technically sound, and do the data support the conclusions?

Reviewer #1: Partly

Reviewer #2: Yes

2. Has the statistical analysis been performed appropriately and rigorously? 

Reviewer #1: Yes

Reviewer #2: Yes

3. Have the authors made all data underlying the findings in their manuscript fully available?

Reviewer #1: Yes

Reviewer #2: Yes

4. Is the manuscript presented in an intelligible fashion and written in standard English?

Reviewer #1: Yes

Reviewer #2: Yes

5. Review Comments to the Author

Reviewer #1: The Authors showed the important role of some inflammatory pathways, in the beneficial effect of exercise in the gout pathology.

The paper is potentially interesting, there are however, some issue that need to be better clarified.

The Authors declare that they have used both male and female mice. However, the results seem not to be divided for the gender. Did the authors obtain the same results for both male and female mice? In the female mice, did the Authors perform the experiments during a specific phase correlated to the menstrual cycle?

This issue should be better discussed since several inflammatory and also pain-associated pathologies, especially those mediated by immune system can be gender dependent.

Interestingly, recent findings showed that voluntary exercise is associated with production of a ketone body beta hydroxybutyrate (BHB) that exerts beneficial effect on emotional behavior through BDNF enhancement (Sleiman et al., ELIFE, 2016), and recently it has been demonstrated that BHB can exert a potent anti-inflammatory and antineuropathic activity (Youm et al., Nat Med, 2015; Qian et al BJP, 2017; Boccella et al., FASEB-J 2019). Moreover, emotional behavior and inflammation have been recently linked (D'Acquisto F, Dialogues Clin Neurosci, 2017 for review). Do the Authors considered those aspect, at least by measuring the glucose and BHB levels in the trained mice? It is clear that this was not the topic of this specific paper, but there could be some cross-talk in these pathways and it would be intriguing to have a "common soil". The question is: did the Authors observed any behavioral changes also in the trained mice?

The quality of the figures could be enhanced

Reviewer #2: Comments to the manuscript ID: PONE-D-20-02077

In this manuscript (Ref. ID: PONE-D-20-02077) entitled “Physical Activity Suppresses Acute Inflammation in a Gout Model by Downregulation of TLR2 on Circulating Neutrophils as well as Inhibition of Serum CXCL1 and is Associated with Decreased Pain and Inflammation in Gout Patients”, Jablonski K. and colleagues show that physical activity at low-moderate extent is able to counteract the pathologic response induced by intra-articular injection of MSU crystals in a mouse model of Gout arthritis. In particular, low-moderate physical activity reduces the degree of ankle swelling and inflammation at site of injection by limiting NF-kB activity and prevents the infiltration of immune cells, mainly macrophages and neutrophils, in the joint. Specifically, the authors report that peripheral neutrophils expressing TLR2 are less abundant in mice subjected to low-moderate exercise compared to control group or high exercise rate group and this effect could be explained by the reduction in serum level of chemokine CXCL1 involved in neutrophils recruitment. Finally, the authors add some data collected from gout patients in which physical activity contributed to improve CRP levels and more in general the pathological score confirming the importance of performing physical activity in both pre-clinical and clinical settings. The manuscript is well organized and the results described are sounded and supported by the experimental data. However, there are some criticisms that need a deep revision. My comments are listed below:

- In their title the authors claim that Physical Activity Suppresses Acute Inflammation. However, animals are subjected to physical activity daily for two weeks before proceeding to MSU injection and, indeed, before inducing an arthritic damage. Therefore, in my opinion the protective effect mediated by physical activity seems to be preventive rather than therapeutic. I would personally suggest to replace the term “suppresses” with “prevents”. In alternative, did the authors consider to test the effect of physical activity on pain and joint inflammation during and/or after arthritis induction?

- For their model characterization, the authors report data showing the degree of ankle swelling, NF-kB activity, immune cells infiltration and pro-inflammatory cytokine production following MSU-injection. However, there are not data showing pain perception in these animals before and after exercise and MSU-injection. Did the author consider to evaluate allodynia and/or hyperalgesia thresholds in these animals in order to check whether physical exercise is able to prevent hyperalgesia typical of arthritis flares?

-In the immunohistochemistry and immunofluorescence paragraphs of methods section, I was wondering why the authors did not perfuse-fixed their mice before collecting tissue for imaging experiments. This could explain the poor quality of the images presented in Figure 1. In this Figure the resolution seems very low and particularly for panels F, G and H the background is too strong making tough the message uptake. This issue should be considered by improving the quality of the image or repeating the staining with a better protocol for tissue fixation.

- In Figure 1 and 3 the authors perform an immunostaining for MPO to evaluate the degree of neutrophil infiltration in the ankle joint, bone marrow and MSU-injected synovium. In my opinion, it should be more appropriate to measure MPO activity rather than its expression meanwhile neutrophil infiltration might be quantified by counting the immunocomplex formation following immunostaining with Ly6G antibody.

-In Figure 4D, it’s not clear how the authors collected peripheral monocytes and neutrophils to stimulate them with LPS and/or MSU. In the text (line 314), they claim that pooled splenocytes were isolated and thereafter monocytes and neutrophils were stimulated meanwhile in the immunoblotting paragraph of method section it is reported that these cells were immunomagnetically separated from whole blood leukocytes. Which was the source of these cells? Spleen or peripheral blood? Moreover, which kind of immunomagnetic beads have been used? Had stimulating growth factors been used to culture these cells prior to stimulate them? Please add some details to clarify this point.

- Are the immunoblots representative of how many samples for each condition? Figure legend reports that n=5 pooled cell lysates were used for condition. This means that only one sample was tested in the immunoblot? Furthermore, a graphical quantification could help readers in the results comprehension.

- Quality of Figure 5A seems very poor and makes not possible to visualize the scale bar used for TLR2-PE-Vio770 quantification. It seems that gate position is cutting in the middle a population of neutrophils not expressing TLR2. Moreover, gating strategy originating neutrophils population should be showed at least as supplemental material. Finally, why did the authors consider to evaluate peripheral cells rather than cells collected from synovial washes?

- For the clinical data, it’s not clear which ones were the inclusion criteria for patient recruitment a part from gout presence and the lack of flares at the moment of visit. Please include a clear statement in the method paragraph to summarize the criteria adopted to include patients in the study (age, BMI, presence of other autoimmune diseases, contemporary assumption of medication at the moment of the study in particular steroids or other anti-inflammatory drugs, etc.,).

-Please add in the method section or in the legends describing imaging experiments how many slides or pictures were taken for each treatment and if quantification was performed in blind.

-In the Discussion section, the authors should include some considerations on the potential mechanism/s of action by which physical activity is producing the effects reported in their results section. In particular, study conducted by F. D'Acquisto and colleagues show that mice housed in an enrich environment and, therefore stimulated to play and indeed perform physical activity, are endowed with an immune system that responds more effectively and faster to infective insults compared to the respective controls. In this manuscript, although the model is reproducing an acute inflammatory situation, physical activity seems to suppress the immune response. Please add some comments on the mechanisms of action supporting the importance of physical exercise in gout patients.

6. PLOS authors have the option to publish the peer review history of their article (what does this mean?). If published, this will include your full peer review and any attached files.

Reviewer #1: No

Reviewer #2: No

---

## [Author Response · Author response to Decision Letter 0]

26 Jul 2020

(Note: please see "Response to reviewers" document included with the re-submission files for the embedded figures referenced here; figures could not be placed in this submission format)

Response to reviewers

Dear Dr. D'Acquisto,

Thank you for including the reviews of our manuscript entitled "Physical Activity Prevents Acute Inflammation in a Gout Model by Downregulation of TLR2 on Circulating Neutrophils as well as Inhibition of Serum CXCL1 and is Associated with Decreased Pain and Inflammation in Gout Patients" (PONE-D-20-02077R1) to which we respectfully submit our revised manuscript. Below, we have addressed the reviewer comments point-by-point. All reviewer comments are in italics and the author(s) responses are included subsequently. We hope to have addressed the informative reviews in sufficient detail to accept our revised manuscript for publication. 

Reviewer #1

Comments to the Authors:

The Authors showed the important role of some inflammatory pathways, in the beneficial effect of exercise in the gout pathology. The paper is potentially interesting, there are however, some issue that need to be better clarified.

Response:

We thank the reviewer for taking the time to critique our submission and are pleased to learn that they feel it is a potentially interesting study. We have clarified the concerns raised specifically below in the point-by-point responses. 

Comments to the Authors:

The Authors declare that they have used both male and female mice. However, the results seem not to be divided for the gender. Did the authors obtain the same results for both male and female mice? In the female mice, did the Authors perform the experiments during a specific phase correlated to the menstrual cycle?

This issue should be better discussed since several inflammatory and also pain-associated pathologies, especially those mediated by immune system can be gender dependent.

Response:

The cohorts and sexes examined in this study were more purposefully predetermined than was described in the previous version of the manuscript. Furthermore, in cases where male and female data were available, they were examined individually and only reported collectively if the data did not demonstrate a significant sex-bias in analysis. This is clarified in the manuscript resubmission (see “track changes” version of the resubmission) and will be outlined below. Male mice were selectively used in this study to investigate inflammatory pathogenesis and disease pathology in order to better reflect the human patient population. Epidemiological data shows that the male sex-bias in the disease reaches disparities of 10:1 when considering pre-menopausal cohorts (1). Consequently, only male mice were injected with MSU for this study. 

Alternatively, in the experiments determining treadmill speeds that correlate with different exercise intensities, male and female BALB/c mice were used. Since no differences were observed between sexes in analysis, the data was reported collectively.

To examine the effects of physical activity in a human gout patient population, physical activity questionnaires and clinical data were analyzed from a cohort of patients with a gout diagnosis (n=30). The patient population consisted of 25 males and 5 females; however, all females were post-menopausal. While the male predilection of gout is 10:1 prior to menopause, this is reduced significantly in post-menopausal cohorts presumably due to the uricosurics effect of female sex hormones (1). Since individual analysis showed no significant correlation to sex in any metric measured, the data from this cohort was reported collectively as well.

When female mice were used in the murine treadmill studies to determine VO2max and to examine the anti-inflammatory effects of moderate exercise without MSU injection, the stage of the menstrual cycle was not determined. In collective analysis and assuming that different stages were randomly represented in the cohort, there was no significant difference to male counterparts, which indicates that sex-bias is not an influencing factor in that dataset. 

References:

1. Singh, J.A., Racial and gender disparities among patients with gout. Curr Rheumatol Rep, 2013. 15(2): p. 307.

Comments to the Authors:

Interestingly, recent findings showed that voluntary exercise is associated with production of a ketone body beta hydroxybutyrate (BHB) that exerts beneficial effect on emotional behavior through BDNF enhancement (Sleiman et al., ELIFE, 2016), and recently it has been demonstrated that BHB can exert a potent anti-inflammatory and antineuropathic activity (Youm et al., Nat Med, 2015; Qian et al BJP, 2017; Boccella et al., FASEB-J 2019). Moreover, emotional behavior and inflammation have been recently linked (D'Acquisto F, Dialogues Clin Neurosci, 2017 for review). Do the Authors considered those aspect, at least by measuring the glucose and BHB levels in the trained mice? It is clear that this was not the topic of this specific paper, but there could be some cross-talk in these pathways and it would be intriguing to have a "common soil". The question is: did the Authors observed any behavioral changes also in the trained mice?

Response:

While we agree that the examination of emotional behavior and the resulting production of biological immunomodulators is beyond the scope of this study, we share the intrigue raised by the reviewer. One important clarification prior to further discussion below is that these mice were not exercising voluntarily. In murine exercise studies, the physical activity regimens can be passively or actively regulated. With passive exercise, an exercise wheel is placed in the cage and the mice are free to use it when they like. This type of experimental set-up requires individualized mouse caging and a system to monitor wheel revolutions to determine distance traveled and speed, which were logistical and financial barriers preventing our adoption of these techniques. On the other hand, in this study, the Exer3/6 treadmill and controller with an electric grid by Columbus Instruments was used. The exercise is not voluntary due to the electrical grid. The comprehensive details of this training are captured in our approved IACUC protocol for this study and will be summarized below for the reviewer’s reference.

Prior to experimental analysis, mice began treadmill training and reached the target speeds for experimental evaluation. Most mice adapt well to walking on the treadmill, and weight loss is not anticipated. However, a baseline weight was obtained and the mice were weighed on a weekly basis. No significant weight loss was observed in any of our mice throughout the study. Before exercising, mice were acclimated to the treadmill for at least 10 minutes with the belt not moving and the electric grid off. Then, the belt was activated at approximately 5 meter/min and exercising began with electric grid still off for an additional 5 minutes, so that the mice could get used to running with minimal distress. After switching the electric grid on, the same speed was maintained for additional 5 minutes and then gradually accelerated to target speeds. The grid provides a stimulus of 163V at 1mA with a shock provided every second, which is a moderate intensity to the animal and is included in our IACUC protocol in adherence to the basic principles of humane endpoints and using removal criteria that meet current standards of animal care (1). A mouse staying on the electric grid for 5 consecutive seconds would be considered exhausted and removed from the treadmill. 

Since no differences in emotional behavior were observed in the mice throughout the course of this study and this specific investigation is beyond the scope of the manuscript, levels of glucose and BHB were not measured/reported and the specific effects of emotional behavior over inflammation were not investigated. The Sleiman et al. manuscript highlighted by the reviewer above is referenced in the revised submission and the absence of significant changes in the emotional behavior of the mice throughout this study is duly noted in the text (see “track changes” version of the resubmission).

Reference:

1. Nemzek, Jean A, Xiao, Hong-Yan, Minard, Anne E, Bolgos, Gerald L, Remick, Daniel G. Humane Endpoints in Shock Research. Shock: January 2004, Volume 21 - Issue 1 p 17-25.

Comments to the Authors:

The quality of the figures could be enhanced

Response:

All of the figures created for submission of this manuscript were made using Adobe Photoshop Version: 21.1.2. All final figure resolutions were greater than or equal to 300 pixels/inch (ppi) in RGB color mode with 16 bit graphics, an enhanced color profile (working RGB: sRGB IEC61966-2.1), and a square pixel aspect ratio. For the histology tissue sections of each figure panel, the slides were digitally scanned using Aperio ImageScope digital analysis version 9.1 software. Since there are no issues with resolution and clarity on our end, we believe this is an artifact of the upload through the journal submission site due to the large file size of each figure. For the resubmission, new figure files were generated and uploaded with modified/enlarged aspect ratios. Additionally, each file was compressed to facilitate uploading to the on-line submission site. We can assure the reviewer that we will work in coordination with the journal to ensure the final publication has images/figures matching the quality on our end.

The final image file sizes are: 5,913 KB for Figure 1 (1,499 KB after compression to reduce file size), 2,873 KB for Figure 2 (283 KB after compression to reduce file size), 3,739 KB for Figure 3 (801 KB after compression to reduce file size), 5,636 KB for Figure 4 (646 KB after compression to reduce file size), 1,996 KB for Figure 5 (143 KB after compression to reduce file size), 4,297 KB for Figure 6 (81 KB after compression to reduce file size), 3,197 KB for Supplemental Figure 1 (504 KB after compression to reduce file size), 3,187 KB for Supplemental Figure 2 (102 KB after compression to reduce file size), 2,660 KB for Supplemental Figure 3 (113 KB after compression to reduce file size).

Reviewer #2

Comments to the Authors:

In this manuscript (Ref. ID: PONE-D-20-02077) entitled “Physical Activity Suppresses Acute Inflammation in a Gout Model by Downregulation of TLR2 on Circulating Neutrophils as well as Inhibition of Serum CXCL1 and is Associated with Decreased Pain and Inflammation in Gout Patients”, Jablonski K. and colleagues show that physical activity at low-moderate extent is able to counteract the pathologic response induced by intra-articular injection of MSU crystals in a mouse model of Gout arthritis. In particular, low-moderate physical activity reduces the degree of ankle swelling and inflammation at site of injection by limiting NF-kB activity and prevents the infiltration of immune cells, mainly macrophages and neutrophils, in the joint. Specifically, the authors report that peripheral neutrophils expressing TLR2 are less abundant in mice subjected to low-moderate exercise compared to control group or high exercise rate group and this effect could be explained by the reduction in serum level of chemokine CXCL1 involved in neutrophils recruitment. Finally, the authors add some data collected from gout patients in which physical activity contributed to improve CRP levels and more in general the pathological score confirming the importance of performing physical activity in both pre-clinical and clinical settings. The manuscript is well organized and the results described are sounded and supported by the experimental data. However, there are some criticisms that need a deep revision. My comments are listed below:

Response:

We thank Reviewer #2 for their time in the consideration of our manuscript and for the positive review provided. Based on the reviewer critiques, the manuscript has been edited as explained below.

Comments to the Authors:

In their title the authors claim that Physical Activity Suppresses Acute Inflammation. However, animals are subjected to physical activity daily for two weeks before proceeding to MSU injection and, indeed, before inducing an arthritic damage. Therefore, in my opinion the protective effect mediated by physical activity seems to be preventive rather than therapeutic. I would personally suggest to replace the term “suppresses” with “prevents”. In alternative, did the authors consider to test the effect of physical activity on pain and joint inflammation during and/or after arthritis induction?

Response:

The point raised by the reviewer is correct regarding the exercise functioning as a preventative intervention more than as a therapeutic one. Considering that the exercise regimen was performed entirely prior to MSU challenge in the data included in the manuscript, the assessment that this is a preventative intervention is accurate. Consequently, the title has been changed from “suppresses” to “prevents”. 

In our complete study, we actually investigated the effects of exercise both before and during/after the induction of gout. In the set of experiments included in the manuscript, the mice were exercised prior to gout induction to examine the influence of exercise frequency and intensity on the inflammatory response. However, since patients do not walk into the clinic before they have gout, we also examined the effects of exercise during/after the induction of gout to determine whether exercise can minimize the pathology associated with flares. In a second set of experiments, we attempted to design a novel animal model where gout would be induced in several rounds with a rest/recovery period in-between, which should better represent the chronic inflammatory milieu that persists in patients with gout. Using the optimal exercise regimen defined in the first set of experiments (11 m/min), we exercised the mice during a recovery period to determine if this has an effect over a subsequent flare of gout. MSU crystals were injected intra-articularly in NFκB-RE-luc mice every 10 days for four cycles. IVIS measurements of localized NFκB activity were taken 14-16 hrs following MSU injection and 4 days later during a rest/recovery period. Longitudinal analysis of localized NFκB activity showed what is representative of a complete resolution in-between each round, as reflected by similar levels at baseline when compared to the rest/recovery periods (Fig. 1). These results indicate that repeated MSU injections alone were insufficient to induce the chronic levels of inflammation required to establish a chronic gout model. 

Since gout patients suffer from elevated uric acid levels, we hypothesized that inducing hyperuricemia in conjunction with repeated MSU challenges would be sufficient to establish a chronic mouse model of gout. In subsequent experiments, subcutaneous injections of uric acid (50 mg/mL uric acid resuspended in 1M sodium hydroxide; 500 µL per injection) were performed daily (approximately 30 mg per mouse) in an attempt to raise systemic levels of uric acid above 6.8 mg/dL, which is the saturation point in serum. Unfortunately, the mice (N = 10) did not tolerate the daily subcutaneous injections due to the lower pH of the solution and all met the early removal criteria of our animal protocol after 7 days due to skin tightening and subdermal inflammation/irritation at the sight of injection (Fig. 2). Two mice faired even more poorly and died of adverse events (exact cause unknown; found dead in the cage) following 2 days of injections. These results indicated that another method of elevating serum uric acid levels would be required to establish a chronic mouse model of gout.

In an additional attempt to induce elevated uric acid levels, we followed a previously established model of hyperuricemia in mice by adding 2% oxonic acid to the drinking water. Oxonic acid is an inhibitor of uricase, which is an enzyme that permits the metabolism and secretion of uric acid in mice. In de-ionized water, oxonic acid (2g) was dissolved in 100 mL diH2O. The solution was heated to 60C for 10 min and stirred gently while keeping covered in order to prevent loss of water via evaporation. The solution was a pH of around 5.5 at this point, but was brought to pH 7.85 ± 0.05 with 8M NaOH, which was the closest to pH 7 with no precipitation observed. The solution was cooled to room temperature and placed in water bottles for the mice. Water was changed weekly for each cage. 

Serum was collected after 4 weeks and analyzed. Compared to mice with normal water, 2% oxonic acid only minimally raised serum uric acid levels and the difference was not statistically significant (Fig. 3). Based on previous studies, this should have elevated uric acid levels more significantly by this point; thus, we decided to explore alternative strategies since this was not observed in our cohort. 

Following a protocol performed by Johnson et al. (published in P.S.E.B.M., vol. 131, 1968) in rats, we contracted a commercially available manufacturer (TestDiet® http://www.testdiet.com/) to prepare a unique formulation of mouse chow containing 5% oxonic acid and 1% uric acid. According to the Johnson et al. study, serum uric acid levels were raised 5-fold in five days and plateaued at a 7-fold increase after one week. Additionally, at necropsy on day 20, they found MSU crystal deposition in the kidneys of this animal model. Although a significant increase in serum urate was detectable, they found that food intake was decreased 20%, weight gain was decreased by 50%, and urine volumes increased 8-9 times when compared to controls. However, these side-effects did not result in any adverse events or intolerance/death in the cohort compared to the control group. Consequently, we closely monitored our mice on this diet and anticipated the potential possibility of some adverse effects and/or unintended consequences. Our rationale to move forward with this approach was based on 3 factors: i) the diet was sustainable to long-term induction of hyperuricemia, ii) MSU crystal deposition was detected in the kidneys in previous studies, and iii) this method was shown to be superior to the administration of oxonic acid or uric acid alone.

NFκB-RE-luc mice were fed the 5% oxonic acid and 1% uric acid TestDiet® formulation of their standard mouse chow and monitored daily via gross examination. After 3 days, lethargy and weight loss were evident and mice did not seem to be eating the food. Mash was made in Petri dishes with water and the mice improved food intake (mash preps were replaced as needed – once to twice daily), but weight loss and lack of ambulatory function was continually observed. At one week, mice met early removal criteria for our animal protocol and had to be removed from the study. Serum was collected to measure uric acid levels and mice were examined by gross necropsy. Just below the stomach in the small intestine, gas bubbles and an inflated tract were observed in all mice (Fig. 4), which may have contributed to decreased food intake. Measurement of serum uric acid levels after 7 days of this 5% oxonic acid and 1% uric acid diet resulted in a significant induction of uric acid from 125 to 215 µM (p ≤ 0.01; Fig. 5). Since human uric acid levels deviate in healthy males compared to those with hyperuricemic gout by approximately 40% (around 5 mg/dL in healthy adults and 8.5 mg/dL in those with gout by estimation), this 42% increase from 125 to 215 µM is comparable to human disease. In conclusion, we were able to induce the target level of hyperuricemia, but the mice did not tolerate the diet. 

Since a chronic model of gout could not be established and complete inflammatory resolution was observed in mice given repeated MSU injections without measures to raise serum uric acid levels, only the acute model data was included in the manuscript. These data are briefly summarized in the discussion section of the revised submission (see track changes version of the revised submission; supplemental figure 5). 

Taken together, we feel that the results of our study show that exercise would be helpful to a patient with gout that is in a recovery period between flares to hopefully prevent or limit future ones, as a preventative measure. While rest and decreased movement/weight may still be recommended to a patient with a flare presenting with a red, painful, and swollen foot, we envision a standardized exercise regimen being prescribed during times of clinical inactivity to help the severity and frequency of future occurrences. 

Comments to the Authors:

For their model characterization, the authors report data showing the degree of ankle swelling, NF-kB activity, immune cells infiltration and pro-inflammatory cytokine production following MSU-injection. However, there are not data showing pain perception in these animals before and after exercise and MSU-injection. Did the author consider to evaluate allodynia and/or hyperalgesia thresholds in these animals in order to check whether physical exercise is able to prevent hyperalgesia typical of arthritis flares?

Response:

As described above, the effects of exercise were examined both before and during/after the induction of gout. However, only the data in the acute gout model is included in the manuscript. In this model, mice were exercised for 2 weeks to be acclimated to the treadmill. Subsequently, MSU challenge occurred immediately following the final exercise session. Thus, the mice were not on the treadmill with inflamed ankles/feet, so an assessment of pain was not possible under those conditions. Furthermore, the experimental endpoint was within 16 hours of the MSU injection, including a 12 hour period of time that was the dark cycle time of the animal research facility. When the animals were observed during this limited window, there were no signs of distress or pain and ambulatory function was normal. Consequently, we do not feel that the evaluation of allodynia or hyperalgesia would have yielded substantive data. In addition, while pain is associated with gouty arthritis (in both mice and humans), the purpose of this study was to examine immunological factors in the context of the disease pathology rather than the associated pain response. 

Comments to the Authors:

In the immunohistochemistry and immunofluorescence paragraphs of methods section, I was wondering why the authors did not perfuse-fixed their mice before collecting tissue for imaging experiments. This could explain the poor quality of the images presented in Figure 1. In this Figure the resolution seems very low and particularly for panels F, G and H the background is too strong making tough the message uptake. This issue should be considered by improving the quality of the image or repeating the staining with a better protocol for tissue fixation.

Response:

The intra-articular joint space injection site of MSU crystals also was the location of the resulting inflammatory infiltrate. While the histopathology showed cells both in the joint space in in surrounding tissue areas, the tissues were not perfused in order to preserve the inflammatory cells in the joint space. Otherwise, cellular architecture would not have been representative of the in vivo pathology. All of the figures created for submission of this manuscript were made using Adobe Photoshop Version: 21.1.2. and the final figure resolutions were greater than or equal to 300 pixels/inch (ppi) in RGB color mode with 16 bit graphics, an enhanced color profile (working RGB: sRGB IEC61966-2.1), and a square pixel aspect ratio. The histology tissue sections of each figure panel were digitally scanned using Aperio ImageScope digital analysis version 9.1 software. Since there are no issues with resolution and clarity on our end, we believe this is an artifact of the upload through the journal submission site due to the large file size of each figure. For the resubmission, new figure files were generated and uploaded as condensed files. We can assure the reviewer that we will work in coordination with the journal to ensure the final publication has images/figures matching the quality on our end.

In this study, all the tissue slides were submitted to the Comparative Pathology & Mouse Phenotyping Shared Resource at The Ohio State University as a contracted service for immunohistochemical (IHC) staining. In this process, antibody optimization testing was first performed to determine a final concentration for staining. With IHC, we typically have higher background with this antibody, as is the case with some IHC antibodies. However, despite some non-specific background staining, the cells positive for staining are clearly distinguishable, which was considered by the staff at the Comparative Pathology & Mouse Phenotyping Shared Resource in the histopathological analysis and recommendation of antibody dilution to use for the final/experimental staining of our slides. Furthermore, since the tissue sections contained bone, decalcification was required in TBD-2 before dissection for paraffin processing and IHC staining. The decalcification process often influences IHC quality and contributes to higher background levels, even with antibody optimization. 

Comments to the Authors:

In Figure 1 and 3 the authors perform an immunostaining for MPO to evaluate the degree of neutrophil infiltration in the ankle joint, bone marrow and MSU-injected synovium. In my opinion, it should be more appropriate to measure MPO activity rather than its expression meanwhile neutrophil infiltration might be quantified by counting the immunocomplex formation following immunostaining with Ly6G antibody.

Response:

Provided the decalcification processing for these tissue sections, the Comparative Pathology & Mouse Phenotyping Shared Resource at The Ohio State University recommended to use an MPO antibody for IHC because commercially-available Ly6G IHC antibodies that would work under these conditions were not able to be identified. Myeloperoxidase (MPO) is most abundantly expressed in neutrophils and suitable to identify these cells by IHC. In addition, MPO is associated with NLRP3 inflammasome activity, which is a known inflammatory mechanism upregulated in gout. 

Immunocomplexes would be defined by an antigen (presumably MSU) and an antibody. While we agree with the reviewer that an investigation of the potential influence of the adaptive immune system and the role that B cells play in the process would indeed be interesting, we feel that the scope of the study was limited to acute inflammation and the innate response in the context of macrophages and neutrophils specifically, which are the primary cells known to be involved in this acute inflammatory response. Consequently, the evaluation of immunocomplex formation is beyond the scope of our study and was not considered in our experiments or manuscript text.

Comments to the Authors:

In Figure 4D, it’s not clear how the authors collected peripheral monocytes and neutrophils to stimulate them with LPS and/or MSU. In the text (line 314), they claim that pooled splenocytes were isolated and thereafter monocytes and neutrophils were stimulated meanwhile in the immunoblotting paragraph of method section it is reported that these cells were immunomagnetically separated from whole blood leukocytes. Which was the source of these cells? Spleen or peripheral blood? Moreover, which kind of immunomagnetic beads have been used? Had stimulating growth factors been used to culture these cells prior to stimulate them? Please add some details to clarify this point.

Response:

The macrophages and neutrophils used in ex vivo LPS/MSU stimulation were derived from whole blood samples. The splenocytes were used for compensation controls (not for experimental analysis) and the text in the manuscript mistakenly had this labeled incorrectly. In the supplemental figure included with the resubmission (please see supplemental Figure 4), the gating strategy is provided and the results from compensation control staining of splenocytes are included. The figure legend (Figure 4) captured the conditions more accurately, but was also edited for clarity in the manuscript resubmission. The objective of this experiment was to assess the potential influence of exercise conditioning on subsequent inflammasome activation in the cells of the peripheral immune system, which was achieved via ex vivo LPS/MSU stimulation. Consequently, whole blood was used and monocytes or neutrophils were isolated by positive selection via immunomagnetic sorting. Since the yields of monocytes and neutrophils from the peripheral blood were expectedly low from individual mice, 5 mice from each experimental condition were pooled so that a sufficient concentration could be analyzed by Western Blot analysis. In the materials and methods, the STEMCELL Technologies immunomagnetic bead kits are indicated for monocyte or neutrophil isolation following manufacturer’s protocol. In lieu of specific growth factors, LPS and MSU were used to prime cells and stimulate inflammasome activation, which is a standard technique for in vitro inflammasome studies. 

Comments to the Authors:

Are the immunoblots representative of how many samples for each condition? Figure legend reports that n=5 pooled cell lysates were used for condition. This means that only one sample was tested in the immunoblot? Furthermore, a graphical quantification could help readers in the results comprehension.

Response:

As detailed above, the protein yields of monocyte or neutrophil isolation by immunomagnetic bead isolation would have been insufficient as individual samples for Western Blot analysis. Therefore, this precluded individual analysis and required a pooled sample to be run. In the revised submission, the Western Blot bands are quantified by densitometry and expressed as an optical density ratio relative to the housekeeping gene (beta-actin). Please refer to the revised submission text and the new version of Figure 4.

Comments to the Authors:

Quality of Figure 5A seems very poor and makes not possible to visualize the scale bar used for TLR2-PE-Vio770 quantification. It seems that gate position is cutting in the middle a population of neutrophils not expressing TLR2. Moreover, gating strategy originating neutrophils population should be showed at least as supplemental material. Finally, why did the authors consider to evaluate peripheral cells rather than cells collected from synovial washes?

Response:

As mentioned above, the images are of high quality and resolution on our end and any issues otherwise are presumably due to a problem with uploading on the journal website. We will work hand-in-hand with the journal editors to ensure that we have significantly improved image quality in the final publication. To aid in uploading in the revised submission, the images were condensed to reduce the file size. Furthermore, we can understand the point raised by the reviewer about the selection of gating position for TLR2-PE-Vio770. To address this concern, we have included a gating strategy figure as a supplement (see supplemental figure 4 in the revised submission), which outlines the compensation control analysis used to make the decisions on the final gating strategy. This confirms that the gates were not arbitrarily selected, despite dissecting the middle of a population of cells. The selection to evaluate peripheral neutrophils rather than those from synovial washes was made because of the question we were answering with this experiment. Specifically, is TLR2 downregulated in neutrophils in the immune system periphery as a result of physical activity, which would potentially provide an explanation why less neutrophils are found in the ankle with exercise pre-conditioning. We already knew by our histopathological observations that less neutrophils were present locally in the ankle; consequently, this experiment was examining what was happening in the immune system periphery that may affect their recruitment to that site of inflammation following MSU crystal injection. 

Comments to the Authors:

For the clinical data, it’s not clear which ones were the inclusion criteria for patient recruitment a part from gout presence and the lack of flares at the moment of visit. Please include a clear statement in the method paragraph to summarize the criteria adopted to include patients in the study (age, BMI, presence of other autoimmune diseases, contemporary assumption of medication at the moment of the study in particular steroids or other anti-inflammatory drugs, etc.,).

Response:

Please refer to the revised submission for a clarification of the inclusion/exclusion criteria used for the selection of gout patients for this study. 

Comments to the Authors:

Please add in the method section or in the legends describing imaging experiments how many slides or pictures were taken for each treatment and if quantification was performed in blind.

Response:

In the methods section in the revised submission, it is indicated that the images were not performed in blinded analysis. Due to the small study team involved in this work, all study members were actively involved in the care of the mice, exercise conditioning, tissue collection, and imaging of the mice, which precluded the ability to effectively blind the data in acquisition/analysis. To mediate this confounding factor in data analysis, the data was generated collectively with all experimental conditions analyzed simultaneously in real-time throughout the study. In addition, the number of pictures for the imaging analysis is included in the figure legends in the revised submission. 

Comments to the Authors:

In the Discussion section, the authors should include some considerations on the potential mechanism/s of action by which physical activity is producing the effects reported in their results section. In particular, study conducted by F. D'Acquisto and colleagues show that mice housed in an enrich environment and, therefore stimulated to play and indeed perform physical activity, are endowed with an immune system that responds more effectively and faster to infective insults compared to the respective controls. In this manuscript, although the model is reproducing an acute inflammatory situation, physical activity seems to suppress the immune response. Please add some comments on the mechanisms of action supporting the importance of physical exercise in gout patients.

Response:

This a very astute and relevant point raised by the reviewer. Consequently, we have added additional text to the discussion in the revised submission that addresses the importance of physical exercise in gout patients. Considered collectively and in a translational context, we feel that the results of our study show that exercise would be helpful to a patient with gout that is in a recovery period between flares to hopefully prevent or limit future ones, as a preventative measure. While rest and decreased movement/weight may still be recommended to a patient with a flare presenting with a red, painful, and swollen foot, we envision a standardized exercise regimen being prescribed during times of clinical inactivity to help the severity and frequency of future occurrences.

---

## [Editor Report · Decision Letter 1]

29 Jul 2020

Physical Activity Prevents Acute Inflammation in a Gout Model by Downregulation of TLR2 on Circulating Neutrophils as well as Inhibition of Serum CXCL1 and is Associated with Decreased Pain and Inflammation in Gout Patients

PONE-D-20-02077R1

Dear Dr. Young,

We’re pleased to inform you that your manuscript has been judged scientifically suitable for publication and will be formally accepted for publication once it meets all outstanding technical requirements.

Kind regards,

Fulvio D'Acquisto, PhD

Academic Editor

PLOS ONE
---

## [Editor Report · Acceptance letter]

10 Sep 2020

PONE-D-20-02077R1 

Physical Activity Prevents Acute Inflammation in a Gout Model by Downregulation of TLR2 on Circulating Neutrophils as well as Inhibition of Serum CXCL1 and is Associated with Decreased Pain and Inflammation in Gout Patients 

Dear Dr. Young:

I'm pleased to inform you that your manuscript has been deemed suitable for publication in PLOS ONE. Congratulations! Your manuscript is now with our production department. 

Kind regards, 

on behalf of

Professor Fulvio D'Acquisto 

Academic Editor

PLOS ONE